# The Relationships between Supply Chain Capability and Shareholder Value Using Financial Performance Indicators

**Seock-Jin Hong [1],* [iD] and Hossein Najmi [2]**

[1]   Department of Marketing, Logistics and Operations Management, G. Brint Ryan College of Business, University of North Texas, Denton, TX 76203, USA

[2]   Department of Information Systems and Operations Management, College of Business, University of Central Oklahoma, Edmond, OK 73034, USA; hnajmi@uco.edu

*   Correspondence: seock.hong@unt.edu

**Abstract:** The purpose of this paper is to explore which financial performance indicators (FPIs) evaluate the level of supply chain capability (SCC) that explicitly touches all of the business functions and processes within and beyond the company. The authors investigated nine FPIs that were selected from the financial statements of 155 companies within nine industries from 2011 to 2017 using Morningstar financial database and Gartner's report. The authors find that suitable FPIs to measure SCC for shareholders' value are return-on-assets (ROA), days-sales-outstanding (DSO), and current ratio (CR). This means that higher ROA, shortened DSO, and an appropriate level of CR could reach a sustainable supply chain. These results will help the industry to avert a major disruption in supply chain processes and activities using suitable financial performance indicators.

**Keywords:** shareholder value; return-on-asset; days-sales-outstanding; current ratio; supply chain capability; sustainable supply chain

---

## 1. Introduction

Researchers in various fields have published numerous articles with diverse research designs that examine the financial impacts on supply chain capability (SCC). Supply chain management (SCM) has a significant impact on a company's financial performance and stock price [1–5]. Supply chain management has a direct impact not only on financial indicators but also the marketing performance of an organization [6], such as increased market share and return on investments [7,8], lower total costs [9], improved customer relations [10], and increased operational efficiency, which includes higher-order fulfillment rates and shorter-order cycle times [9]. It also influences competitive advantage [6,11], and the supply chain strategy has a central position in creating shareholder value (SHV) [12] to assure sustainable supply chain [13].

Supply chain management has been defined to explicitly recognize the strategic coordination between trading partners to improve an individual organization's performance and to improve the whole supply chain [11,14]. Within leading companies, the SCC hinges on the health and well-being of the critical ecosystems within and around them, including people, the planet, and the partnerships formed to deliver customer solutions [15]. Higher supply chain capability has a positive effect on a firm's performance regarding increased market share, shareholder value, revenue growth, fixed capital efficiency, operating cost reductions, and working capital efficiency [6,12] (see Figure 1). However, despite the increased attention paid to financial performance and SCM, relatively few studies utilize a wide range of financial indicators to cover company-wide financial performance ratios to evaluate supply chain capability. Many studies attempt to analyze working capital efficiency using cash-to-cash

(C2C) cycle time, or one or two financial indicators, which limits access to company-wide supply chain processes and activities [6,16–18].

Supply chain management revolves around coordination, cooperation, and especially collaboration [4] among inter-organizational and business partners that are linked by the flow of materials, money, and information [19]. The complex relationships up and downstream make it difficult to acquire related data for the entire supply chain and SHV. To address this research gap, we provide a general framework to evaluate joint supply chain efforts to improve shareholder value using common SCC related financial performance indicators (FPIs) beyond C2C and categories of financial ratios to analyze company-wide health and try to find a competitive differentiator that influences shareholder value. The SCC is decisively important for operational efficiency, working capital management, and, ultimately, the bottom line, whereas a CEO ought to be fully engaged [11]. Therefore, the purpose of this research is to find the relationship between shareholder values and supply chain capability using companies' financial statements from 2011 to 2017. The remainder of this paper is organized into five sections. Section 2 presents a review of the literature on conceptual frameworks with several hypotheses that address the characteristics of SCC regarding FPIs. Section 3 discusses the data collection process, research methodology, and results. Section 4 contains the discussion and implications, and Section 5 concludes.

## 2. Literature Review and Hypotheses

Economic value added (EVA) contributes to creating shareholder value [20] and gradually substitutes cost and profit objective functions to design a supply chain network [3,21]. Shareholders' perspectives always inform managerial decisions because every company must do its best to keep shareholders and bondholders happy [22]. The ultimate purpose of the company is to maximize SHV for the long-term worth of the business to its owners [12]. The supply chain strategy has a central position in SHV creation and is the main source of competitive advantage [12]. The basic drivers to enhance SHV are revenue growth [12,15], operating cost reductions, fixed capital efficiency, working capital efficiency [12], earnings before interest, taxes, depreciation, and amortization (EBITDA), earnings per share (EPS) [12,22], and economic value added [20,23]. We have selected year-over-year changes in a firm's revenue and EPS as our measures of shareholder value. These measures are often the first number that companies report to investors in their quarterly earnings call [12,15,22] because these measures provide evidence of value created by the firm to its shareholders.

Financial metrics (or ratios) are a window into a company's financial statements [22]. One important factor in business is an ongoing performance measurement [18]. However, previous literature has applied FPIs separately and not covered an extensive analysis of its supply chain capabilities and activities using comprehensive FPIs. We categorize the FPIs into three different areas that managers and other stakeholders in a business typically use to analyze the company's SCC. Based on previous research, we classify 13 FPIs into three different groups [22]—profitability, operational efficiency and liquidity to measure SCC as well as SHV—as displayed in Table 1. We make four assumptions based on nine FPIs that have a positive relationship with supply chain performance.

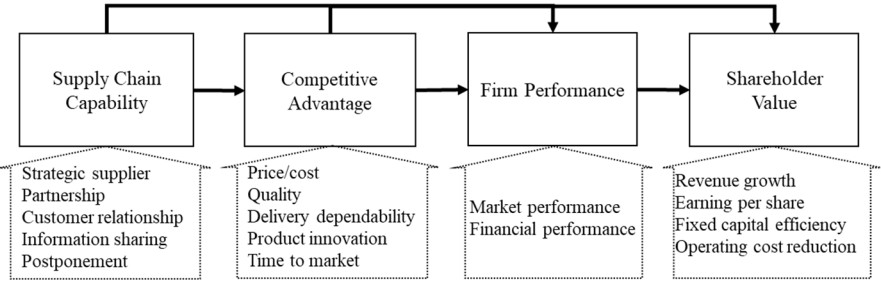

**Figure 1.** Link supply chain capability and shareholder value. Source: [6,12] and authors' elaboration.

**Table 1.** Financial Performance Indicators.

| Category [1] | Financial Performance Indicators (FPIs) | Definitions | Influence on SC Capability [3] | Sources [2] |
|---|---|---|---|---|
| Profitability | Return-on-sales (ROS) | Net profit/revenues | Yes | [11,24–30] |
| | Gross profit margin (GP) | Gross profit/revenues | Yes | |
| | Return-on-assets (ROA) | Net profit/assets | Yes | |
| | Operating profit | Operating profit/revenue | NC | N/A |
| | Return on equity (ROE) | Net profit/shareholders' equity | NC | N/A |
| Operational Efficiency | Days in inventory (DII) | Average inventory/(COGS/day) | Yes | [11,12,16,17,30–32] |
| | Inventory turnover | Cost of goods sold/Average inventory | Yes | |
| | Days-sales-outstanding (DSO) | Ending accounts receivable/(revenue/day) | Yes | |
| | Days payable outstanding (DPO) | Ending account payables/(revenue/day) | Yes | |
| | Asset turnover | Sales/Average total assets | Yes | |
| | C2C (cash-to-cash) cycle | DII+DSO-DPO | Yes | |
| | C2C YoY | Percentage change of C2C year over year | Yes | |
| | Property, plant, and equipment turnover (PPE) | Revenue/PPE | NC | N/A |
| Liquidity (Solvency) | Current ratio (CR) | Current assets/current liabilities | Yes | [29,33] |
| | Quick ratio (QR) | (Current asset-inventory)/current liability | NC | |
| Leverage | Debt-to-equity | Total liabilities/shareholders' equity | NC | N/A |
| | Interest coverage | Operating profit/annual interest charges | NC | |
| Shareholders Value (SHV) | Revenue changes YoY (Year over Year) | % change of revenue to the same period of the previous year | Yes | [12,23] |
| | EPS (Earning per share) in dollars | (Net income - Dividends on preferred stock) / Average outstanding shares | Yes | |

[1] Based on [22]. [2] References just in case that FPIs have an influence on supply chain performance. [3] NC: No Commented on literature.

Profitability ratios, such as net profit margin (ROS; return-on-sales), gross profit margin (GP), and return-on-assets (ROA) evaluate a company's ability to generate profits through making sales and controlling expenses [22]. The ROS tells a company how much of every sale they keep after everything else has been paid for including people, vendors, lenders, the government, etcetera [22]. The ROS is net profit divided by revenue. The GP margin shows the basic profitability of the product or service and is calculated by gross profit divided by revenue. The GP indicates a potential problem for a company; when the GP is heading downward or becoming negative, it is assumed that the company has been considerably discounting products and is under severe price pressure [22]. The ROA shows how effectively the company uses its assets to generate profits; the equation is net profit over total assets. Most of the literature shows that SCM (green [34] and sustainable SCM [35] ) has a positive impact on a firm's performance in areas like net profit margin [24,25,27], gross profit margin [12,30], and return-on-assets [11,15,26,28]. This discussion forms the basis of the following hypothesis with three indicators (ROS, GP, and ROA) together:

**H1:** *Profitability (ROS, GP, and ROA) has a positive relationship with shareholder value.*

Supply chain practices could improve cash flows and reduce the C2C cycle time [12,16,17,31,36], which would help free up cash and working capital to be invested in other products, better processes, and better financial performance [11,37]. Cash flow is a key indicator of a company's financial health, along with profitability and shareholders' equity [22,38]. The C2C covers the end-to-end of the supply chain and gives a certain diagnostic view based on inventory. Cash-to-cash is a critical performance measure of operational performance and has an impact on supply chain practices [12,16,17,36], but is not a one-size-fits-all strategy and managers in smaller firms should pay close attention to their C2C [38]. The C2C cycle time is defined as the sum of the day-sales-outstanding (DSO), plus the day-in-inventory (DII), minus the days payable outstanding (DPO), that is C2C = DSO + DII − DPO. The C2C is a critical performance measure and was also selected as the measure that has the greatest impact on supply chain practices because it shows the direct financial benefits of SCM [17,31] with improving the revenues of a company by 3% to 6% [16]. Wang's [39] research results showed that

reducing the C2C improved the operating performance of a firm. Prior research has found a significant negative relationship between profitability and the measures of working capital management, such as C2C [40,41].

Moreover, prior findings also indicate a significant negative correlation between C2C and measures of firm performance such as net sales and total assets [12,16,17,31,36]. It suggested that companies could increase profits by correctly managing the C2C cycle time and keeping the components of C2C at an optimum level. The C2C metric is an important measure because it bridges across inbound material activities with suppliers through manufacturing operations and outbound sales activities with customers [16]. The C2C increases the visibility of decision variables, increases the optimization of decisions in the supply chain, reduces sub-optimization of the financial decision within firms, and aids supplier decision-making by eliminating the uncertainty of customer actions [17]. The focus on managing C2C is the premise that a reduction in the cash conversion cycle time will lead to financial and operational improvements. However, within the supply chain, a leading player, likely located downstream, could take the initiative to shorten C2C significantly [42]. The strongest player in the supply chain could finance weak suppliers and customers [18]. This assumption could reduce the attractiveness of the product to the customers as the cost of goods increases. The operating cash flows are sensitive to declining sales and earnings [43]. From this alternate point of view, investors should focus on cash flows from mobilizing inventory (inventory turnover [15]), receiving investments, and using its assets efficiently to increase sales (asset turnover). Thus, operational efficiency encompasses not only C2C but also inventory turnover, asset turnover, and changes of the C2C year over year. With SCC, a firm should see increased operational efficiency in terms of increased asset and inventory turnover while reducing C2C and change of C2C year over year. Thus, we hypothesize:

**H2:** *There is a negative relationship between SHV and operational efficiency (asset turnover, DSO, DII, DPO, and change of C2C year over year).*

Liquidity ratios measure the short-term ability to pay debt obligations. They consist of the current ratio (CR) [33], the quick ratio, and the cash ratio. Liquidity ratios are closely connected to cash management in a supply chain [29,33]. Credit solvency is one of the essential pillars of financial status that can provide the necessary capital to a supply chain network [23]. Liquidity and solvency ratios measure the ability of the company to pay its obligations over the short and long runs. We focus more on short-run ability with CR. The current ratio measures a company's current assets against its current liabilities. The current assets are those that can be converted into cash in less than a year; this figure (Figure 2) normally includes accounts receivables and inventory as well as cash [22]. Thus, we posit:

**H3:** *There is a positive relationship between supply chain capability and liquidity (CR).*

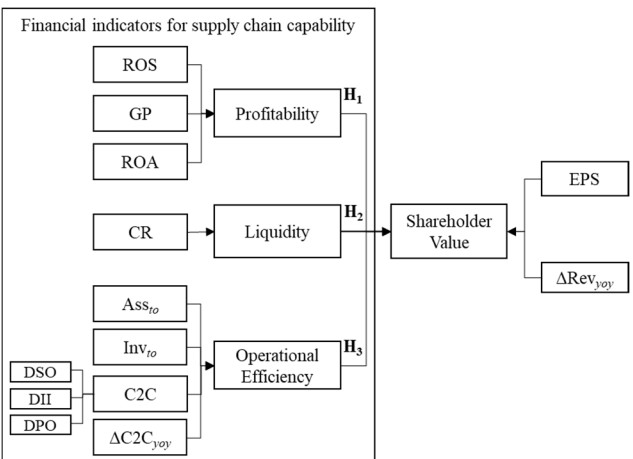

**Figure 2.** Research model: The relationship of supply chain capability and shareholder value. Source: Authors' elaboration.

## 3. Research Methodology

We use the following procedure to develop our theory: (1) specify FPIs, (2) review literature on FPIs and supply chain capability, (3) collect data for FPIs and sample companies, (4) apply ordinary least square (OLS) regression analysis with all possible variables, (5) find an appropriate model assessing the variance inflation factor (VIF) for the severity of multicollinearity, normality, and homoscedastic test, (6) apply 1000 bootstrap replications if the tests are not significant statistically, and (7) verify the hypotheses using a significant level of dependent variables.

### 3.1. Data Collection

The Morningstar®Investment Research Center offers comprehensive financial data for investors, academics, and practitioners. First, we collected financial data from 311 companies within 18 industries based on 2015 data. Among the 311 companies, we chose 157 companies with more than $1 billion in revenue, and among the top 20 companies within each industry (see Appendix A Table A1). From 157 companies, 42 companies are the supply chain top-performers based on Gartner's report from 2011 to 2017 [15]. Two companies, Dell and Inditex, were excluded because of limited data. We classified the selected 155 companies, which are 40 top-performers as a group (Gr. 1) and 115 companies as a group (Gr. 2) to compare the SCC. Among these 40 top-performing companies, 17 companies were chosen for seven consecutive years, one for six years, two for five years, two for four years, three for three years, five for two years, and ten companies for only one year. Both academics and practitioners refer to Gartner's [44] Supply Chain Top 25 for SCC (Appendix A Table A1). Most of the companies are located in North America (50%), especially the United States (45.8%), Europe (25%), and Asia (24%) including Japan (16%). Eighty companies (51.6%) are over $10 U.S. billion dollars, including 40 of Gartner's top performers (Table 2).

### 3.2. Analysis

In this section, the authors break down the two different parts. The first part focuses on the leading supply chain companies; the authors offer a leading companies' ranking by applying data envelopment analysis (DEA) based on the ranking. The newly pooled ranking gives comprehensive information on which companies have supply chain capabilities. The second part applies OLS to find SCC related variables that have a relationship with SHV. For this analysis, we have two groups of companies: excellence SCC companies (Gr. 1) and non-excellence companies (Gr. 2).

Leading supply chain companies: The data envelopment analysis (DEA) can be used to measure efficiency using multiple input and output variables [45]. This research applies the DEA method to the preference voting method developed by Cook and Kress (CK) [46]. The CK model has been widely used as a decision measurement technique to balance the shortcomings of traditional techniques that are based on preference voting, in which the ranked voting data can be changed depending on the weight [47]. The authors applied the CK model to determine the excellence SCC companies based on the supply chain top 25 from 2011 to 2017, as determined by Gartner Incorporated [15,44]. Using Gartner's ranking data for seven years, the authors tried to measure the comprehensive ranking by applying the preference voting method. (DEA CK is calculated using $Max\ Z_i = Max\ \sum_{j=1}^{k} W_j V_{ij}(\varepsilon)$, where $Z_i$ is the long-term supply chain excellence for company $i$ and $V_{ij}$ is the frequency of $j$th place rank of company $i$ ($i = 1, \dots, m, j = 1, \dots, k$) subject to $\sum_{j=1}^{k} W_i V_{ij} \leq 1\ \forall_i$, $W_j - W_{j+1} \geq d(j, \varepsilon)$, $\forall_j$, where $W_j$ is the coefficient of supply chain excellence and $W_k \geq d(k, \varepsilon)$ for $k$ = last ranking company. $d(\cdot, \varepsilon)$ is a discrimination intensity function with non-negative and non-decreasing $\varepsilon$. The equation of $W_j - W_{j+1} \geq d(j, \varepsilon)$ means that the weight value of $W_j$ of $V_j$ should be larger that the weight value $W_{j+1}$ of $V_{j+1}$. This paper applies $W_j \geq W_{j+1} + 0.0050$ between the 1st to 10th rank, $W_j \geq W_{j+1} + 0.0025$ between 11th to 25th, $W_j \geq W_{j+1} + 0.0010$ between 26th to 35th, and $W_j \geq W_{j+1} + 0.0005$ for 36th to 42nd (see Appendix A Table A2), applying strong ordering [47,48]. The result gives two 42 × 42 matrices for each company's ranking results ($W_j V_{ij}$) and a weight value of each company for each

rank ($W_j$ for all of $i$). Based on the calculation, the authors reached the final ranking of SCC companies (the first column of Appendix A Table A2 based on the third column). Based on this analysis, we divided Group 1 into two sub-groups; one group included those over 0.500 of DEA ranking ([Gr 1 *] top 13 among 42 excellence SCC companies with two excluded companies, such as Inditex and Dell, Appendix A Table A2), and the other group included the remaining companies to apply analysis in the next section further. The Gr. 1 * companies were in the top 25 for seven years in a row and were in the top 10 for at least two years from 2011 to 2017 except H&M for 2011.

**Table 2.** Geographical Location and Annual Sales of Sample Companies.

| Geographical Location | | | | Total Company Annual Sales | | |
|---|---|---|---|---|---|---|
| | | | | Annual Sales | # | % |
| North America | United States | 71 | 78 | 50% | <1 billion U.S. Dollars | 2 | 1% |
| | Canada | 5 | | | 1–5 billion U.S. Dollars | 51 | 33% |
| | Mexico | 2 | | | 5–10 billion U.S. Dollars | 22 | 14% |
| Europe | United Kingdom | 12 | 38 | 25% | 10–20 billion U.S. Dollars | 24 | 15% |
| | Germany | 10 | | | 20–30 billion U.S. Dollars | 12 | 8% |
| | France, Sweden | 4 each | | | 30–40 billion U.S. Dollars | 5 | 3% |
| | Switzerland | 3 | | | >40 billion U.S. Dollars | 39 | 25% |
| | Netherlands | 2 | | | | | |
| | Finland, Italy, Spain | 1 each | | | | | |
| Asia | Japan | 25 | 37 | 24% | | | |
| | China | 5 | | | | | |
| | Korea | 3 | | | | | |
| | Hong Kong, Philippines, Thailand, Turkey | 1 each | | | | | |
| South America | Chile | 1 | | 1% | | | |
| Oceania | New Zealand | 1 | | 1% | | | |
| | Total | 155 | | 100% | | 155 | 100% |

The relationship between SCC and SHV: We conducted OLS based on the previous section. We developed two models—the dynamic model (Equation (1)) and the pooled model (Equation (2))to see how the financial performance indicator-related supply chain capability has a relationship with the shareholder value for excellence SCC companies (Gr 1 * and Gr 1) and non-excellence companies (Gr 2). Equations (1) and (2) are composed of independent variables ([IVs] with selected FPIs related to SCC from previous research, $SHV_{it}$, for Gr. 1, and 2 or Gr. 1 *, 1, and 2. The C2C-related variables are average DSO, DII, DPO, average changes of C2C from 2011 to 2016 as IVs. We take the revenue changes and earning per share in dollars as DVs, $SHV_{it}$ for Equation (1) and $SHV_i$ for Equation (2).

The Dynamic Model with FPIs is $Y_{it} = \beta \mathbf{x}_{it-1} + \varepsilon_{it-1}$, where $\mathbf{x}_{it-1}$ is the independent variable vector. This model assumes that all of the usual OLS assumptions have not been violated, and the effect of any given X and Y is constant across observations with no interaction in X. This model reflects carry-over activities in two consecutive years (see Figure 3) and gives a short-term perspective. This model takes into account the time change effect and does not focus exclusively on a separate period. To cope with the long-term point of view, the dynamic model incorporates carry-over activities into the model [49]. The idiosyncratic differences across years are of interest in dynamic or global changes in the supply chain.

$$SHV_{it} = \beta_0 + \beta_1(ROS)_{it-1} + \beta_2(GP)_{it-1} + \beta_3(ROA)_{it-1} + \beta_4(Ass_{to})_{it-1} + \beta_5 ln(Inv_{to})_{it-1} +$$
$$\beta_6\left(\Delta C2C\_yoy\right)_{it-1} + \beta_7(CR)_{it-1} + \beta_8 ln(DSO)_{it-1} + \beta_9 ln(DII)_{it-1} + \beta_{10} ln(DPO)_{it-1} + \varepsilon_{it-1} \tag{1}$$

where *i* is companies, *t* is a year from 2012 to 2016 when *i* = 1, … *m*, *t* = 1, … ,*n*. When we analyze two groups, Gr. 1 in the fourth column of Appendix A Table A1, and Gr. 2 in the fifth column of Appendix A Table A1. When we have three Groups, Gr. 1 includes excellence SCC companies below 0.5000 of DEA CK results in Appendix A Table A2 (29 companies), and 1 * includes super excellence SCC companies that exceed 0.5001 (13 companies).

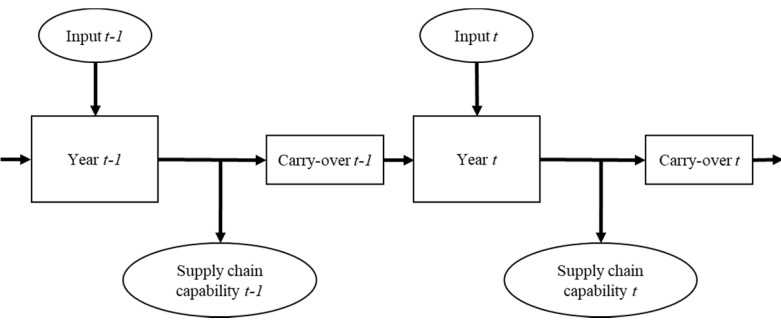

**Figure 3.** Sharing resources between consecutive years and supply chain capability for the dynamic model (Equation (1)).

The pooled model with FPIs is $Y_i = \beta x_i + \varepsilon_i$, where $x_i$ is the independent variable vector with the same assumption as a dynamic model. This single-period model, made by pooling the 5-year data on average (compound annual growth rate for the percentage data), enabled us to measure the relationship without the fluctuating performances of combined good and bad years and long-term horizons. Economic uncertainty refers to macroeconomic, financial, and market conditions that either partially or totally play a role in the supply chain. The supply chain foundation addresses the importance of relationships based on trust [4] and long-term orientation [50–52]. Instead of splitting raw data year over year, the pooled data (combined data) smooths out economic uncertainty and gives the long-term horizon. It may be more appropriate to generalize to a population by pooling data over time to test the long-term relationship and a wide range of collaboration. Pooled data increases the degree of freedom through a financial benefit of increased heterogeneity.

$$SHV_i = \beta_0 + \beta_1(ROS)_i + \beta_2(GP)_i + \beta_3(ROA)_i + \beta_4\left(Ass_{to\_avg}\right)_i + \beta_5 ln\left(Inv_{to\_avg}\right)_i +$$
$$\beta_6(\Delta C2C\_yoy\_a)_i + \beta_7(CR)_i + \beta_8 ln(DSO)_i + \beta_9 ln(DII)_i + \beta_{10} ln(DPO)_i + \varepsilon_i \tag{2}$$

where *i* is companies when *i* = 1, … *m*. All of the IVs are average or geometric means $\left(= \left[\prod_{t=1}^n \theta_{it}\right]^{1/n}\right)$ for percentage data such as the ROS, GP, and ROA values from *n* = 2011 to 2015. To conduct OLS, we tested the models' validity with the multicollinearity, normality and homoscedastic test using VIF, Doornik–Hansen omnibus (D-H) test, and Breusch–Pagan (B-P) test. The mean VIF of the dynamic and pooled models is 2.49 and 1.94 that is under 2.5 (see Table 5). A VIF detects the multicollinearity of IVs (predictors) in the regression analysis. The VIFs are usually calculated by $[1/(1 - R_i^2)]$ with *i*th independent variables. In some studies, a VIF above 10 indicates a high correlation, and less than 10 is acceptable [53]. Five is the maximum level of VIF, and some conservatively use 2.5 [54]. We use 2.5 as the maximum level of VIF for analysis of the hypotheses. After conducting the VIF test, we get nine variables out of 11; two variables were removed including inventory turnover ($Inv_{to}$ or $Inv_{to\_avg}$), which is highly correlated with DII and C2C, which is highly correlated with DSO, DII, and DPO. The D-H and B-P tests show a rejection of the null hypothesis, which means the estimation of models are not normally distributed and homoscedastic of variance. Therefore, we conduct 1000 bootstrap replications to estimate $\hat{\theta}_i$ where *i* = 1, 2, … ., *k* (for this research *k* = 1000) from the observed value of $\hat{\theta}$.

## 4. Discussion and Implications

Table 3 shows the descriptive statistics for Gr. 1 (excellence SCC companies), Gr. 2 (non-excellence SCE companies), and the total mean, and tests of equality between groups based on Wilk's lambda for dynamic and pooled models. Table 4 presents the information for Gr. 1 * (super excellence SCC; top 13 companies from Gr. 1 based on DEA CK in Sections 3.1 and 3.2), Gr. 1, and Gr. 2 with the F-test (ANOVA), and post hoc test (Bonferroni) results. Based on the equality, F-, and Bonferroni tests, the dynamic model is more sensitive on a group-by-group basis than the pooled model, which means that the dynamic model shows more differences and reflects actual changes. The profitability is the order of Gr. 1 > Gr. 2, and Gr. 1 * > Gr. 1 > Gr. 2 in terms of ROS, GP, and ROA. However, ROS and ROA are only significant statistically. The efficiency of C2C related variables shows that the order of Gr. 1 < Gr. 2 is significant only for DII and Gr. 1 * < Gr. 2 < Gr. 1 for changes in C2C changes ($\Delta C2C_{yoy}$), C2C, DSO, DII, DPO, and only DII is significant. Asset turnover and inventory turnover are the order of Gr. 1 > Gr. 2. Gr. 1 > Gr. 2 > Gr. 1 * for asset turnover, and Gr. 1 * > Gr. 2 > Gr. 1 for inventory turnover without significance. For SHV, the changes in revenue are not statistically significant group by group, regarding EPS Gr. 1 > Gr. 2 and Gr. 1 * > Gr. 1 > Gr. 2 with significance. As for liquidity, it is not significant statistically for the difference. The pooled model with three different groups is presented on the right-handed side in Table 4. For the pooled model for Equation (1) in Table 4, none of the FPIs are significant and are in nearly the same order with the dynamic model. The orders of each variable are the same as in Table 4. However, only two variables (ROA and asset turnover) from the category of profitability and operational efficiency are statistically significant. From the descriptive statistics (Tables 3 and 4), we get significant information on SCC using FPIs. The excellence SCC companies are more profitable especially on ROS and ROA, operational efficiency with less C2C cycle time with shortening DSO and DII, and higher SHV with EPS.

**Table 3.** Group Statistics for Gr. 1 and Gr. 2 of Dynamic and Pooled Models.

| Category | FPIs | Statistics for the Dynamic Model (Equation (1)) | | | | Statistics for the Pooled Model (Equation (2)) | | | |
|---|---|---|---|---|---|---|---|---|---|
| | | Gr. 1, Mean ($n$ = 92) | Gr. 2, Mean ($n$ = 455) | Total, Mean ($n$ = 547) [1] | Test of Equality [2] | Gr. 1, Mean ($n$ = 33) | Gr. 2, Mean ($n$ = 109) | Total, Mean ($n$ = 142) [3] | Test of Equality [2] |
| Profitability | ROS | 0.1273 | 0.0664 | 0.0766 | 0.000 *** | 0.0941 | 0.0601 | 0.0680 | 0.058 |
| | GP | 0.4508 | 0.4100 | 0.4168 | 0.066 | 0.4906 | 0.4173 | 0.4343 | 0.084 |
| | ROA | 0.1169 | 0.0679 | 0.0761 | 0.000 *** | 0.0875 | 0.0637 | 0.0692 | 0.072 |
| Operational efficiency | $\Delta C2C_{yoy}$ | −0.2615 | −0.1961 | −0.2071 | 0.883 | −0.2064 | -0.9396 | −0.7692 | 0.460 |
| | C2C | 37.627 | 53.136 | 50.494 | 0.044 * | 47.647 | 54.592 | 52.981 | 0.623 |
| | DSO | 37.925 | 46.231 | 44.594 | 0.012 * | 44.061 | 43.703 | 43.786 | 0.957 |
| | DII | 58.644 | 80.229 | 76.908 | 0.001 *** | 62.478 | 85.824 | 80.410 | 0.040* |
| | DPO | 61.048 | 73.324 | 71.008 | 0.023 * | 64.237 | 76.331 | 73.527 | 0.253 |
| | $Ass_{to}$ | 0.0113 | 0.0111 | 0.0111 | 0.777 | 0.0506 | 0.0260 | 0.0317 | 0.338 |
| | $Inv_{to}$ | 16.060 | 12.679 | 13.247 | 0.311 | 13.767 | 13.335 | 13.435 | 0.940 |
| Liquidity | CR | 1.6983 | 1.8390 | 1.8154 | 0.246 | 1.6467 | 1.8354 | 1.7916 | 0.339 |
| SHV | $\Delta Rev_{yoy}$ | 0.0607 | 0.0847 | 0.0806 | 0.516 | 0.0172 | 0.0705 | 0.0581 | 0.036 * |
| | EPS | 9.7322 | 3.4725 | 4.5253 | 0.001 *** | 7.2913 | 3.3127 | 4.2373 | 0.189 |

[1] Total n = 766 with 219 missing data. [2] Test of equality of Group means (based on Wilk's lambda statistics). [3] Total n = 155 with 13 missing data. *** Significant at $\alpha$ = 0.001; ** Significant at $\alpha$ = 0.01; * Significant at $\alpha$ = 0.05.

Table 5 shows the results of OLS and the 1000 bootstrap replications for the dynamic and pooled models using revenue changes as a DV that is an endogenous variable. We apply bootstrapping to avoid random data influence based on the D-H and B-P tests. For the analysis of the dynamic and pooled models, only the pooled model shows meaningful information. The dynamic model shows the actual changes and the pooled model shows the long-term-based result with SHV. Therefore, the DPO and CR have a high positive relationship, and ROS, ROA, DSO has a negative relationship with SHV for the pooled model.

**Table 4.** Group Statistics for the Dynamic and Pooled Model (Equation (1)) of Gr. 1 *, Gr. 1, and Gr. 2.

| Category | FPIs | Statistics for the Dynamic Model (Equation (1)) | | | | | | | Statistics for the Pooled Model (Equation (2)) | | | | | | |
|---|---|---|---|---|---|---|---|---|---|---|---|---|---|---|---|
| | | Gr. 1 * | Gr. 1 | Gr. 2 | Total | Gr. 1 * & 1 | Gr. 1 & 2 | Gr. 1 * & 2 | Gr. 1 * | Gr. 1 | Gr. 2 | Total | Gr. 1 * & 1 | Gr. 1 & 2 | Gr. 1 * & 2 |
| Profitability | ROS | 0.1277 | 0.1174 | 0.0661 | 0.0748 | - | 0.0513 *** | 0.0616 *** | 0.1320 | 0.0849 | 0.0587 | 0.0686 | - | - | 0.0740 * |
| | GP | 0.4791 | 0.4269 | 0.4098 | 0.4158 | - | - | 0.0692 * | 0.4704 | 0.4690 | 0.4138 | 0.4279 | - | - | - |
| | ROA | 0.1241 | 0.1110 | 0.0689 | 0.0764 | - | 0.0421 *** | 0.0552 *** | 0.1246 | 0.0758 | 0.0651 | 0.0717 | - | - | |
| Operational efficiency | $\Delta C2C_{yoy}$ ($\Delta C2C_{yoy\_a}$) | −0.1809 | −0.3482 | −0.1954 | −0.2082 | - | - | - | −0.0303 | −0.25932 | −0.9505 | −0.7637 | - | - | - |
| | C2C | 28.123 | 49.345 | 51.408 | 49.621 | - | - | - | 25.4033 | 59.935 | 48.479 | 49.906 | - | - | - |
| | DSO | 27.009 | 47.344 | 45.569 | 44.334 | −20.335 ** | - | −18.559 *** | 25.5860 | 46.731 | 43.649 | 43.003 | - | - | - |
| | DII | 54.792 | 67.5067 | 80.921 | 78.037 | - | - | −26.128 ** | 49.1057 | 74.218 | 82.280 | 78.539 | - | - | - |
| | DPO | 53.209 | 65.5043 | 73.301 | 71.278 | - | - | −20.091 * | 48.1620 | 71.202 | 74.428 | 72.043 | - | - | - |
| | $Ass_{to}$ | 0.0119 | 0.0365 | 0.0319 | 0.0309 | - | - | - | 25.6946 | 7.9225 | 13.266 | 13.182 | - | - | - |
| | $Inv_{to}$ | 13.889 | 9.3027 | 13.222 | 12.938 | - | - | - | 0.0116 | 0.1148 | 0.0120 | 0.0305 | −0.103 *** | - | - |
| Liquidity | CR | 1.5644 | 1.6919 | 1.8279 | 1.7976 | - | - | - | 1.5853 | 1.7114 | 1.8318 | 1.7910 | - | - | - |
| SHV | $\Delta Rev_{yoy}$ ($\Delta Rev$) | 0.0760 | 0.0725 | 0.0764 | 0.0761 | - | - | - | 0.0561 | 0.0153 | 0.0773 | 0.0645 | - | - | - |
| | EPS | 14.235 | 4.1480 | 3.2979 | 4.1599 | 10.087 *** | - | 10.937 *** | 13.479 | 3.7396 | 3.1863 | 4.0831 | - | - | - |

*** Significant at $\alpha = 0.001$; ** Significant at $\alpha = 0.01$; * Significant at $\alpha = 0.05$.

Based on an analysis, ROS and GP have a negative relationship with SHV, and ROA has a positive relationship (support partially H1). Therefore, we find a positive relationship between profitability and SCC. The second hypothesis links operational efficiency, including C2C-related variables and asset and inventory turnover to SHV. Most previous research show that the C2C has negative relationships on SCC [6,12,16–18,31,37,39,55] and inventory periods [56]. However, our interesting research results show that the changes of C2C do not have any relationship to SHV; only DSO has a negative relationship when we have three categories of SCCs, as evidenced by the excellence SCC companies having a very short period of DSO. These results partially support the negative relationship between operational efficiency with the changes in revenue (SHV: $\Delta\text{Rev}_{yoy}$) in the long-term horizon (H2). This research focuses more on liquidity using the current ratio for the ability of payment in the short run than solvency with long-run ability. It shows a positive relationship on SHV as previous research mentioned there was a relationship. Thus, H3 is supported. According to Reference [12], SCC creates SHV through the long-run worth of the business to its owners and investors.

Supply chain management is a complex, technology-driven discipline that reaches across functions, business processes, and corporate boundaries [2,4,11,20]. However, most research addresses the SCM problems in an isolated manner and focuses on data from a certain year without analyzing comprehensive financial performance indicators or only focuses on working capital-C2C. The top executives in a company tend to focus on financial performance measures, such as sales, profits, stock prices, and costs of capital to improve SCC [11], and on performance measures aligned with supply chain objectives across multiple firms [20]. Even though delivering SCM is important to financial outcomes, the previous research [18,57] focuses on short-term operative improvements due to complex networks of interrelated activities. The SCM has been the focus of growing research interest in improving profits for all parties involved in the integrated flow of products (or materials), information, and money across multiple companies. Therefore, our research has focused on a wide range of FPIs that influence supply chain capability and has taken into account short-term (dynamic model) and long-term (pooled model) points of view with the same period of data to improve the future financial performance of a particular firm and the supply chain as a whole.

The effectiveness of SCM is reducing DSO [11], C2C [41]; ensuring profitability, growth, and competitiveness [23]; and increasing in ROA [11]. However, shortening the C2C time cycle could also be achieved through delaying payment to suppliers and reducing accounts receivables from customers without any further effort on operational efficiencies, instead of eliminating days of inventory and frequent deliveries with small lot sizes. Shortening the payments to suppliers creates liquidity pressures for other companies in the supply chain. Within the supply chain, a leading player, likely located downstream, could take the initiative to shorten C2C significantly [42]. However, DII could be one of the best metrics to measure SCC instead of C2C [18]. The reduction of the inventory holding period has a positive effect on the C2C cycle time, both from an individual firm as well as a collaborative supply chain viewpoint. This implies that the supply chain parties should seek ways to reduce each member's inventory holding period [56,58]. Such inventory reduction efforts can be realized using other alternatives such as operations technology, right batch sizes, just-in-time approaches, build-to-order production, vendor-managed inventory concepts [59], and enhanced end-to-end relationships through the sharing of information [12,51,56].

Supply chain capability is decisively important for operational efficiency, working capital management, and ultimately, the bottom line [11]. Operational efficiency has been central to some of the greatest success stories in recent business history, including Wal-Mart, Toyota, and Dell [55]. Operation efficiency can lead to high-revenue growth, lower inventory using cross-docking and responsive purchasing and distributing of goods, lower prices, and increased profits, but operational performance is difficult to realize [55]. To improve operational performance, a firm must use supply chain practices [6,37,55], change the business culture [55], and introduce six sigma [11] and lean techniques [60]. Specifying goals for improvements in these areas requires knowing where the company currently stands. Previous research shows that C2C could explain operational efficiency.

Several studies proved that shortening C2C means reducing the terms of credit for the receiver and delaying payment to suppliers. However, if the company tries to reduce the C2C by shortening the DSO, it could reach the effectiveness of SCM in the long run for shareholders' value.

Supply chain processes interface with multiple suppliers and customers and trigger collaborative activities in the long-term; these activities should be based on trust to minimize transaction costs [4,28,61]. A combined supplier-customer EVA analysis enables us to determine how collaborative action leads to the attainment of supply chain outcomes [20]. The pooled model is used to examine the interdependence of supply chain activities through the combined data of all FPIs within five years. This research shows that sustainable long-term finance outcomes could be possible through the positive relationship between customer and supplier, reducing operating expenses, and increasing profitability [20]. Shortening DSO gives way to the positive relationship in the long run between supplier and buyer, which is a source of competitive advantage and generates great returns.

**Table 5.** Results of the OLS and Bootstrap of the Dynamic and Pooled Models (Dependent variable is ΔRev_yoy).

| Category | FPIs | Dynamic Model | | Pooled Model | | Results |
|---|---|---|---|---|---|---|
| | | OLS | Bootstrap (1000) | OLS | Bootstrap (1000) | |
| | | Coeff. (*sig.*) [1] | | Coeff. (*sig.*) [1] | | |
| H$_1$: Profitability | ROS | −0.4580 (*0.106*) | −0.4580 (*0.212*) | −0.9204 (*0.041* *) | −0.9204 (*0.061*) | Supported partially |
| | GP | −0.0976 (*0.349*) | −0.0976 (*0.401*) | −0.1888 (*0. 036* *) | −0.1888 (*0.056*) | |
| | ROA | 0.7201 (*0.068*) | 0.7201 (*0.052*) | 1.0044 (*0.000* ***) | 1.0044 (*0.005* **) | |
| | ΔC2C$_{yoy}$ | 0.0071 (*0.046* *) | 0.0071 (*0.502*) | 0.0002 (*0.914*) | 0.0002 (*0.964*) | |
| H$_2$: Operational efficiency | ln(DSO) | −0.0485 (*0.009* **) | −0.0485 (*0.064*) | −0.0279 (*0.014* *) | −0.0279 (*0.011* *) | Supported weakly |
| | ln(DII) | −0.0026 (*0.898*) | −0.0026 (*0.871*) | 0.0261 (*0.070*) | 0.0261 (*0.166*) | |
| | ln(DPO) | 0.0578 (*0.056*) | 0.0578 (*0.077*) | 0.0414 (*0.038* *) | 0.0414 (*0.095*) | |
| | Ass$_{to}$ | −1.5193 (*0.658*) | −1.5193 (*0.619*) | −0.2548 (*0.003* **) | −0.2548 (*0.680*) | |
| | ln(Inv$_{to}$) | - | - | - | - | |
| H$_3$: Liquidity | CR | 0.0188 (*0.220*) | 0.0188 (*0.231*) | 0.0286 (*0.015* *) | 0.0286 (*0.007* **) | Supported |
| Constant | | 0.02734 | 0.02734 | −0.0908 | −0.0908 | |
| Number of obs | | 544 | | 137 | | |
| Mean VIF | | 2.49 | - | 1.94 | - | |
| Doornik–Hansen test (*sig.*) H$_0$: Normality | | $\chi^2$ = 24810 (0.000 ***) | - | $\chi^2$ = 8650 (0.000 ***) | - | - |
| Breusch–Pagan test (*sig.*) H$_0$: Homogenous | | $\chi^2$ = 12.63 (0.262) | - | $\chi^2$ = 28.01 (0.000 ***) | - | |
| F (or $\chi^2$)-value (*sig.*) | | *F-value* = 2.26 (*0.006* **) | $\chi^2$ = 23.38 (*0.0054* **) | *F-value* = 7.47 (*0.000* ***) | $\chi^2$ = 27.31 (*0.001* **) | |
| Adj R$^2$ | | 0.0205 | 0.0205 | 0.2998 | 0.2998 | |

[1] Coefficients of OLS and bootstrap are the same but the significant levels are different. *** Significant at α = 0.001; ** Significant at α = 0.01; * Significant at α = 0.05.

## 5. Conclusions with Limitations and Future Research Directions

Among the financial indicators we used in this study to express supply chain capability, days-sales-outstanding (DSO) is one of the most important metrics to measure comprehensive supply chain capability in the category of operational efficiency, return-on-assets in profitability, and current ratio in liquidity in the long-term for shareholder value. In particular, super excellence SCC companies show very short DSO and DII. This means that supply chain benefits share not only themselves but also others by shortening payment times to reduce the financial pressure to suppliers. Relying on C2C to control supply chain management as shown on previous research [17,36,62], it possibly weakens their control supply chain capability and sustainability [13,63] beyond the company in the long-term, and makes it difficult to ensure that their suppliers are operating in a financially sustainable fashion [18]. From a sustainable supply chain perspective, if suppliers have weaker credit ratings and thus pay higher interest rates than their customers pay, collaborative supply chain finance could not be possible [64]. Supply chain management deals with several decision variables regarding warehousing dollars, transportation, and optimal inventory levels [11,23] as well as buy-or-make decisions, distribution centers, and other common measures used for global optimization instead of local optimization [18].

Many companies measure only what they can easily access [11] to see the factors that affect supply chain processes and activities. Supply chain management has become a complicated set of activities that involves many business functions and processes, along with competitive differentiators [11]. Financial performance is one of the essential pillars that provide the necessary capital to supply chain networks [23]. Therefore, we use a wide range of financial performance indicators to help measure the supply chain capability, ensuring that both customers' expectations and stakeholders' benefit. Other values of this research include a holistic approach to reach a collaborative supply chain to find supply chain capability along with financial sustainability.

There are several important areas for future research to measure the supply chain capability using financial performance indicators, such as extending it to the end-to-end supply chain network, buyer-supplier finance, the effect of the firm's size and organizational/corporate culture, which have an important role to SCC and shareholders' value.

**Author Contributions:** S.-J.H. designed and developed the study model including conceptualization, methodology, validation, writing—review and editing, and H.N. collected data, software, and visualization. All authors have read and agreed to the published version of the manuscript.

**Funding:** This research received no external funding.

**Acknowledgments:** The authors wish to thank the anonymous referees for their valuable comments.

**Conflicts of Interest:** The authors declare no conflict of interest.

## Appendix A

**Table A1.** Industry, Top 25 Companies (Gr. 1) and Comparison Companies (Gr. 2).

| # | Industry | Number | Top 25 Companies (Gr. 1) | Comparison Companies (Gr. 2) |
|---|---|---|---|---|
| 1 | Discount and retail store | 18 | Wal-Mart Stores, Inc., TESCO, Amazon.com, Nike, Home Depot | Costco Wholesale, Target, Dollar General, Dollar Tree, Dollarama, Lawson, Burlington Stores, Don Quijote Holdings, B&M European Value Retail, Distribuidora Internacional De, Pricesmart, Big Lots, Grupo Gigante SAB de CV |
| 2 | Restaurant | 18 | McDonald's Starbucks | Compass Group, Yum Brands, Chipotle Mexican Grill, Restaurant Brands International, Yum China Holdings, Darden Restaurants, Aramark, Whitbread, Domino's Pizza, Panera Bread, Minor International, Cracker Barrel Old Country Store, Jollibee Foods, Jack In The Box, The Wendy's, Texas Roadhouse. |
| 3 | Household and personal products | 18 | Procter & Gamble (P&G), Unilever, L'Oreal, Colgate-Palmolive, Kimberly-Clark | Reckitt Benckiser, Henkel, The Estee Lauder, Kao, Newell Brands, Svenska Cellulosa, Beiersdorf, Clorox, Coty, Unicharm, Church & Dwight, Shiseido, Hengan International. |
| 4 | Apparel manufacturing | 15 | H&M, Inditex [1] | VF, Under Armour, Hanesbrands, Ralph Lauren, PVH, Michael Kors, Gildan Activewear, Carter's Hugo Boss, Columbia Sportswear, UniFirst, Kate Spade, boohoo.com, Wacoal, G-III Apparel |
| 5 | Beverage and Foods | 13 | Coca-Cola, Pepsi Co., Diageo, Kraft Foods, Nestlé | Monster Beverage, Dr. Pepper Snapple Group, Arca Continental SAB de CV, Embotelladora Andina SA, ITO EN, Britvic, Cott, Refresco Group |
| 6 | Consumer Electronics, communication systems, software, data storage, and Electronics industry | 27 | Apple, Samsung Electronics, Hewlett Packard, Lenovo Group, Research In Motion, Nokia, Cisco Systems, Intel, Qualcomm, Schneider Electric, Microsoft, IBM, Seagate Technology, Dell [1] | Sony, Panasonic., Kyocera, Sharp, LG Display, Harman International Industrie, Electrolux, Alps Electric, Haier Electronics Group, Arcelik, Casio Computer, De'Longhi, Dometic Group, Skyworth Digital Holdings, GoPro |

**Table A1.** *Cont.*

| # | Industry | Number | Top 25 Companies (Gr. 1) | Comparison Companies (Gr. 2) |
|---|----------|--------|--------------------------|------------------------------|
| 7 | Pharmaceutical | 19 | Johnson & Johnson, GlaxoSmithKline | Pfizer, Novartis, Merck, Sanofi, AbbVie, Bristol-Myers Squibb, Bayer, Eli Lilly, AstraZeneca, Astellas Pharma, Otsuka Holdings, Chugai Pharmaceutical, Daiichi Sankyo, Ono Pharmaceutical, Kyowa Hakko Kirin, CSPC Pharmaceutical Group, Santen Pharmaceutical |
| 8 | Automotive | 14 | Toyota, Ford, BMW, | Daimler, Volkswagen, General Motors, Honda Motor., Nissan Motor., Tesla Motors., Audi, Renault, Hyundai Motor, Fiat Chrysler Automobiles, Suzuki Motor |
| 9 | Others | 11 | Cummins, Caterpillar, 3M, BASF | Compagnie de Saint-Gobain, LafargeHolcim, Tempur Sealy International, Fletcher Building, Ricoh, La-Z-Boy, Steelcase |
| | | 155 | 40 (2) | 115 |

[1] Excluded from the analysis because of data acquisition limit in the Morning Star database.

**Table A2.** Leading supply chain companies based on Gartner's Top 25 companies from 2011 to 2017.

| DEA CK Ranking | Company | SCC Index [1] | Difference of SCE | Gartner's Supply Chain Ranking [2] | | | | | | | |
|---|---|---|---|---|---|---|---|---|---|---|---|
| | | | | # of Top 25 | 2017 | 2016 | 2015 | 2014 | 2013 | 2012 | 2011 |
| 1 | Apple | 1.0000 | | 7 | 1 [3] | 1 [3] | 1 [3] | 1 | 1 | 1 | 1 |
| 2 | Amazon.com | 0.9350 | 0.0650 | 7 | 3 [3] | 3 | 1 | 3 | 3 | 2 | 5 |
| 3 | McDonald's | 0.9300 | 0.0050 | 7 | 2 | 2 | 2 | 2 | 2 | 3 | 8 |
| 4 | P&G | 0.8650 | 0.0650 | 7 | 5 [3] | 5 [3] | 5 [3] | 5 | 6 | 5 | 3 |
| 5 | Unilever | 0.8521 | 0.0129 | 7 | 1 | 1 | 3 | 4 | 4 | 10 | 15 |
| 6 | Cisco Systems | 0.8100 | 0.0421 | 7 | 4 | 7 | 6 | 7 | 7 | 8 | 6 |
| 7 | Intel | 0.7996 | 0.0104 | 7 | 6 | 4 | 4 | 8 | 5 | 7 | 16 |
| 8 | Inditex | 0.7901 | 0.0095 | 7 | 3 | 6 | 5 | 11 | 12 | 15 | 19 |
| 9 | Coca-Cola | 0.6862 | 0.1039 | 7 | 14 | 9 | 11 | 10 | 9 | 6 | 11 |
| 10 | Samsung Electronics | 0.6818 | 0.0044 | 7 | 25 | 8 | 8 | 6 | 8 | 13 | 10 |
| 11 | Colgate-Palmolive | 0.6680 | 0.0138 | 7 | 9 | 13 | 9 | 9 | 10 | 11 | 13 |
| 12 | H&M | 0.6041 | 0.0639 | 6 | 5 | 5 | 7 | 13 | 17 | 17 | |
| 13 | Wal-Mart Stores | 0.5481 | 0.0560 | 7 | 18 | 16 | 13 | 14 | 13 | 9 | 7 |
| 14 | Nike | 0.4612 | 0.0869 | 7 | 8 | 11 | 10 | 12 | 14 | 14 | 20 |
| 15 | PepsiCo | 0.4355 | 0.0257 | 7 | 11 | 15 | 15 | 15 | 16 | 12 | 9 |
| 16 | Starbucks | 0.4310 | 0.0045 | 7 | 10 | 12 | 12 | 17 | 15 | 16 | 22 |
| 17 | 3M | 0.3932 | 0.0378 | 7 | 12 | 14 | 14 | 18 | 19 | 21 | 24 |
| 18 | Nestlé | 0.3589 | 0.0343 | 7 | 7 | 10 | 17 | 25 | 21 | 18 | 18 |
| 19 | Dell | 0.3573 | 0.0016 | 3 | | | | | 11 | 4 | 2 |
| 20 | Johnson & Johnson | 0.3412 | 0.0161 | 7 | 13 | 21 | 21 | 22 | 25 | 22 | 21 |
| 21 | HP | 0.2933 | 0.0479 | 4 | 19 | 17 | | | | 24 | 17 |
| 22 | Kimberly-Clark | 0.2724 | 0.0209 | 5 | 21 | 24 | 20 | 21 | | 25 | |
| 23 | Cummins | 0.2485 | 0.0239 | 4 | | | 23 | 24 | 23 | 23 | |
| 24 | Lenovo Group | 0.2439 | 0.0046 | 5 | 24 | 25 | 18 | 16 | 20 | | |
| 25 | Caterpillar | 0.2109 | 0.0330 | 3 | | | | 23 | 18 | 20 | |
| 25 | L'Oréal | 0.2109 | 0.0000 | 3 | 20 | 19 | 22 | | | | |
| 27 | Qualcomm | 0.2081 | 0.0028 | 3 | | | 19 | 19 | 24 | | |
| 28 | Research In Motion | 0.2019 | 0.0062 | 2 | | | | | | 19 | 4 |
| 29 | Schneider Electric | 0.1564 | 0.0455 | 2 | 17 | 18 | | | | | |
| 30 | BASF | 0.1536 | 0.0028 | 2 | 16 | 20 | | | | | |
| 30 | Seagate Technology | 0.1536 | 0.0000 | 2 | | | 16 | 20 | | | |
| 32 | BMW | 0.1312 | 0.0224 | 2 | 22 | 22 | | | | | |
| 33 | Microsoft | 0.0889 | 0.0423 | 1 | | | | | | | 12 |
| 34 | IBM | 0.0836 | 0.0053 | 1 | | | | | | | 14 |

**Table A2.** *Cont.*

| DEA CK Ranking | Company | SCC Index [1] | Difference of SCE | Gartner's Supply Chain Ranking [2] | | | | | | | |
|---|---|---|---|---|---|---|---|---|---|---|---|
| | | | | # of Top 25 | 2017 | 2016 | 2015 | 2014 | 2013 | 2012 | 2011 |
| 35 | Nokia | 0.0809 | 0.0027 | 1 | 15 | | | | | | |
| 36 | Ford Motors | 0.0656 | 0.0153 | 1 | | | | | 22 | | |
| 37 | GlaxoSmithKline | 0.0628 | 0.0028 | 1 | | 23 | | | | | |
| 37 | TESCO | 0.0628 | 0.0000 | 1 | | | | | | | 23 |
| 37 | Diageo | 0.0628 | 0.0000 | 1 | 23 | | | | | | |
| 40 | Toyota Motors | 0.0600 | 0.0028 | 1 | | | 24 | | | | |
| 41 | Home Depo | 0.0572 | 0.0028 | 1 | | | 25 | | | | |
| 41 | Kraft Foods | 0.0572 | 0.0000 | 1 | | | | | | | 25 |

[1] SCC index based on DEA CK, which is a relative capability, not an absolute capability. [2] Data source: Gartner (2018-a). [3] Masters: If the companies place in the top 5 rankings for at least 7 out of the past 10 years (Source: https://www.gartner.com). When the companies are nominated as a Master, the ranking does not appear on the list. Therefore, the authors put the last ranking of the company.

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
