# Peer review of "The Relationships between Supply Chain Capability and Shareholder Value Using Financial Performance Indicators"

_sustainability, doi:10.3390/su12083130_

Round 1

Reviewer 1 Report

Comments

  1. The time frame of the performed analysis should be included both in the abstract and in the introduction.
  2. The time frame or frames of the performed analysis should be clearly mentioned
  3. Please comment more in the conclusions part on the added value of the paper, on the practical implications of the findings, in relation with the sustainability issue.
  4. Linguistic errors have been made in the paper, I strongly recommend that the paper should be proofread. Some sentences are unclear/should be reformulated/several affirmations are redundant:

Here are just some examples:

Line 28: […] has a direct impact not only on a financial indicator (I would suggest using the plural instead) but also on the marketing performance of an organization […], such as increased market share and return on investments ( ROI cannot be referred as examples of marketing performance, please reformulate)

Line 49: To address this research gap, we provide a general framework to evaluate joint supply chain efforts to improve shareholder value using common SCC related financial performance indicators (FPIs) beyond C2C and categories of financial ratios to analyze company-wide health and try to find a competitive differentiator that influences shareholder value.  (A reformulation is needed in order to show more simply what is the aim of the research)

Line 53: The SCC is decisively important for operational efficiency, working capital management, and, ultimately, the bottom line, whereas a CEO ought to be fully engaged (A reformulation is needed)

Line 55: this research is to find (I would suggest using “the purpose/scope of the research”)

Line 63: Shareholders’ perspectives always inform managerial decisions ( Reformulate, the sentence is not clearly written)

Line 78: We categorize the FPIs into three different areas that managers and other stakeholders in a business typically use to analyze the company’s SCC: profitability, operational efficiency, and liquidity […]. Based on previous research, we classify 13 FPIs into three different groups—profitability, operational efficiency and liquidity (it was already mentioned previously, the affirmation is redundant)

Line 93: The ROS tells a company how much of every sale they keep (Reformulate).

The reference to Figure 1 should appear in text.

In table 1, what does NC stands for?

Line 174: Among these 40 top-performing companies, 18 companies were chosen for six consecutive years, one for six years, two for five years, two for four years, three for three years, five for two years, and ten companies for only one year (They don’t sum up! 40 or 41?)

Line 177: we analyzed 155 companies within nine industries (the affirmation is redundant)

Line 179: The annual  sales of these companies exceed $1 billion U.S. dollars (the affirmation is redundant)

Line 193: This research applies the DEA method to the preference voting method (explain more clearly this paragraph)

Line 203: 42 excellence SCC companies or 40???

Author Response

  1. The time frame of the performed analysis should be included both in the abstract and in the introduction.

(Answer) Thank you for the constructive suggestions and comments. Your comments are right and appropriate. The authors try to clear the sentences as follow.

(Abstract, line #14 to 16)) The authors investigated nine FPIs that were selected from the financial statements of 155 companies within nine industries from 2011 to 2017 using Morningstar financial database and Gartner’s report.

(Introduction, line #55-56) Therefore, this research is to find the relationship between shareholder values and supply chain capability using companies’ financial statements from 2011 to 2017.

  1. The time frame or frames of the performed analysis should be clearly mentioned

(Answer) Please refer to above my responses above #1 and line # 197, 224.

The time frame is from 2011 to 2017 (Line #14-16, 55-56, 197) for whole paper and preference voting method and from 2012 to 2016 for relationship analysis.

  1. Please comment more in the conclusions part on the added value of the paper, on the practical implications of the findings, in relation with the sustainability issue.

(Answer) We tried to reinforce the section 4 and 5.

  1. Linguistic errors have been made in the paper, I strongly recommend that the paper should be proofread. Some sentences are unclear/should be reformulated/several affirmations are redundant:

(Answer) Thank you for your invaluable suggestions and comments. We did proofread and changed appropriately.

Here are just some examples:

  1. Line 28: […] has a direct impact not only on a financial indicator (I would suggest using the plural instead) but also on the marketing performance of an organization […], such as increased market share and return on investments ( ROI cannot be referred as examples of marketing performance, please reformulate)

(Answer) We made change.

  1. Line 49: To address this research gap, we provide a general framework to evaluate joint supply chain efforts to improve shareholder value using common SCC related financial performance indicators (FPIs) beyond C2C and categories of financial ratios to analyze company-wide health and try to find a competitive differentiator that influences shareholder value.  (A reformulation is needed in order to show more simply what is the aim of the research)

(Answer) Thank you for the constructive suggestions and comments. Your comments are right and appropriate. However, we decided to keep above sentences to focus on research gap. Because we have a sentence to highlight the research focus (line #55-56)

(line #55-56) Therefore, the purpose of this research is to find the relationship between shareholder values and supply chain capability using companies’ financial statements from 2011 to 2017.

  1. Line 53: The SCC is decisively important for operational efficiency, working capital management, and, ultimately, the bottom line, whereas a CEO ought to be fully engaged (A reformulation is needed)

(Answer) Thank you for your invaluable suggestions and comments. As you can see, the phrase “the bottom line, whereas a CEO ought to be fully engaged” is cited from Slone et al., 2007. We would like to keep this words because it explain why the bottom line (profit) is important to CEO and companies’ sustainability. 

Slone, R. E., Mentzer, J. T., & Dittmann, J. P. Are you the weakest link in your company’s supply chain? Harvard Business Review 2007, 85 (September), 116-127.

  1. Line 55: this research is to find (I would suggest using “the purpose/scope of the research”)

(Answer) We made change.

  1. Line 63: Shareholders’ perspectives always inform managerial decisions ( Reformulate, the sentence is not clearly written)

(Answer) Many thanks for your comment. We decided to keep the sentence as the citation (Berman and Knight, 2013) because it is very clear to explain of how shareholders’ perspectives are important.

Berman, K., & Knight, J. Financial intelligence: A manager’s guide to knowing what the numbers really mean. Harvard Business Press. 2013.

  1. Line 78: We categorize the FPIs into three different areas that managers and other stakeholders in a business typically use to analyze the company’s SCC: profitability, operational efficiency, and liquidity […]. Based on previous research, we classify 13 FPIs into three different groups—profitability, operational efficiency and liquidity (it was already mentioned previously, the affirmation is redundant)

(Answer) We appreciate your feedback and deleted the redundancy.

  1. Line 93: The ROS tells a company how much of every sale they keep (Reformulate).

(Answer) Many thanks for your comment. We decided to keep the sentence as the citation (Berman and Knight, 2013) because it is very clear to explain on ROS.

Berman, K., & Knight, J. Financial intelligence: A manager’s guide to knowing what the numbers really mean. Harvard Business Press. 2013.

  1. The reference to Figure 1 should appear in text.

(Answer). We appreciate your feedback. It is already mentioned at the line 40.

  1. In table 1, what does NC stands for?

(Answer) Many thanks for your comment. NC: No commented on literature (Insert the explanation on the line 90.

  1. Line 174: Among these 40 top-performing companies, 18 companies were chosen for six consecutive years, one for six years, two for five years, two for four years, three for three years, five for two years, and ten companies for only one year (They don’t sum up! 40 or 41?)

(Answer) We made changes as follows.

“Among these 40 top-performing companies, 17 companies were chosen for seven consecutive years, one for six years, two for five years, two for four years, three for three years, five for two years, and ten companies for only one year.” (17+1+2+2+3+5+10=40)

  1. Line 177: we analyzed 155 companies within nine industries (the affirmation is redundant)

(Answer). Thank you. We have deleted the redundancy.

  1. Line 179: The annual sales of these companies exceed $1 billion U.S. dollars (the affirmation is redundant)

(Answer). Thank you. We have deleted the redundancy.

  1. Line 193: This research applies the DEA method to the preference voting method (explain more clearly this paragraph)

(Answer) Many thanks for your comment. If you could see the line 200 “…the preference voting method1.”, we added the explanation to footnote #1 (please see following explanation).

DEA CK is calculated using  , where is the long-term supply chain excellence for company i and  is the frequency of jth place rank of company i (i=1,…,m, j=1,…,k) subject to

  1. The time frame of the performed analysis should be included both in the abstract and in the introduction.

(Answer) Thank you for the constructive suggestions and comments. Your comments are right and appropriate. The authors try to clear the sentences as follow.

(Abstract, line #14 to 16)) The authors investigated nine FPIs that were selected from the financial statements of 155 companies within nine industries from 2011 to 2017 using Morningstar financial database and Gartner’s report.

(Introduction, line #55-56) Therefore, this research is to find the relationship between shareholder values and supply chain capability using companies’ financial statements from 2011 to 2017.

  1. The time frame or frames of the performed analysis should be clearly mentioned

(Answer) Please refer to above my responses above #1 and line # 197, 224.

The time frame is from 2011 to 2017 (Line #14-16, 55-56, 197) for whole paper and preference voting method and from 2012 to 2016 for relationship analysis.

  1. Please comment more in the conclusions part on the added value of the paper, on the practical implications of the findings, in relation with the sustainability issue.

(Answer) We tried to reinforce the section 4 and 5.

  1. Linguistic errors have been made in the paper, I strongly recommend that the paper should be proofread. Some sentences are unclear/should be reformulated/several affirmations are redundant:

(Answer) Thank you for your invaluable suggestions and comments. We did proofread and changed appropriately.

Here are just some examples:

  1. Line 28: […] has a direct impact not only on a financial indicator (I would suggest using the plural instead) but also on the marketing performance of an organization […], such as increased market share and return on investments ( ROI cannot be referred as examples of marketing performance, please reformulate)

(Answer) We made change.

  1. Line 49: To address this research gap, we provide a general framework to evaluate joint supply chain efforts to improve shareholder value using common SCC related financial performance indicators (FPIs) beyond C2C and categories of financial ratios to analyze company-wide health and try to find a competitive differentiator that influences shareholder value.  (A reformulation is needed in order to show more simply what is the aim of the research)

(Answer) Thank you for the constructive suggestions and comments. Your comments are right and appropriate. However, we decided to keep above sentences to focus on research gap. Because we have a sentence to highlight the research focus (line #55-56)

(line #55-56) Therefore, the purpose of this research is to find the relationship between shareholder values and supply chain capability using companies’ financial statements from 2011 to 2017.

  1. Line 53: The SCC is decisively important for operational efficiency, working capital management, and, ultimately, the bottom line, whereas a CEO ought to be fully engaged (A reformulation is needed)

(Answer) Thank you for your invaluable suggestions and comments. As you can see, the phrase “the bottom line, whereas a CEO ought to be fully engaged” is cited from Slone et al., 2007. We would like to keep this words because it explain why the bottom line (profit) is important to CEO and companies’ sustainability. 

Slone, R. E., Mentzer, J. T., & Dittmann, J. P. Are you the weakest link in your company’s supply chain? Harvard Business Review 2007, 85 (September), 116-127.

  1. Line 55: this research is to find (I would suggest using “the purpose/scope of the research”)

(Answer) We made change.

  1. Line 63: Shareholders’ perspectives always inform managerial decisions ( Reformulate, the sentence is not clearly written)

(Answer) Many thanks for your comment. We decided to keep the sentence as the citation (Berman and Knight, 2013) because it is very clear to explain of how shareholders’ perspectives are important.

Berman, K., & Knight, J. Financial intelligence: A manager’s guide to knowing what the numbers really mean. Harvard Business Press. 2013.

  1. Line 78: We categorize the FPIs into three different areas that managers and other stakeholders in a business typically use to analyze the company’s SCC: profitability, operational efficiency, and liquidity […]. Based on previous research, we classify 13 FPIs into three different groups—profitability, operational efficiency and liquidity (it was already mentioned previously, the affirmation is redundant)

(Answer) We appreciate your feedback and deleted the redundancy.

  1. Line 93: The ROS tells a company how much of every sale they keep (Reformulate).

(Answer) Many thanks for your comment. We decided to keep the sentence as the citation (Berman and Knight, 2013) because it is very clear to explain on ROS.

Berman, K., & Knight, J. Financial intelligence: A manager’s guide to knowing what the numbers really mean. Harvard Business Press. 2013.

  1. The reference to Figure 1 should appear in text.

(Answer). We appreciate your feedback. It is already mentioned at the line 40.

  1. In table 1, what does NC stands for?

(Answer) Many thanks for your comment. NC: No commented on literature (Insert the explanation on the line 90.

  1. Line 174: Among these 40 top-performing companies, 18 companies were chosen for six consecutive years, one for six years, two for five years, two for four years, three for three years, five for two years, and ten companies for only one year (They don’t sum up! 40 or 41?)

(Answer) We made changes as follows.

“Among these 40 top-performing companies, 17 companies were chosen for seven consecutive years, one for six years, two for five years, two for four years, three for three years, five for two years, and ten companies for only one year.” (17+1+2+2+3+5+10=40)

  1. Line 177: we analyzed 155 companies within nine industries (the affirmation is redundant)

(Answer). Thank you. We have deleted the redundancy.

  1. Line 179: The annual sales of these companies exceed $1 billion U.S. dollars (the affirmation is redundant)

(Answer). Thank you. We have deleted the redundancy.

  1. Line 193: This research applies the DEA method to the preference voting method (explain more clearly this paragraph)

(Answer) Many thanks for your comment. If you could see the line 200 “…the preference voting method1.”, we added the explanation to footnote #1 (please see following explanation).

DEA CK is calculated using  , where is the long-term supply chain excellence for company i and  is the frequency of jth place rank of company i (i=1,…,m, j=1,…,k) subject to  , , where is the coefficient of supply chain excellence and  for k = last ranking company.  is a discrimination intensity function with non-negative and non-decreasing . The equation of  means that the weight value of  of  should be larger that the weight value  of . This paper applies  between the 1st to 10th rank,  between 11th to 25th,  between 26th to 35th, and  for 36th to 42nd (see Appendix Table 2), applying strong ordering [27, 43]. The result gives two 42 x 42 matrices for each company’s ranking results ( and a weight value of each company for each rank ( for all of i).

  1. Line 203: 42 excellence SCC companies or 40???

(Answer) We made changes as follows.

“…42 excellence SCC companies with two excluded companies, such as Inditex and Dell..”

  , , where is the coefficient of supply chain excellence and  for k = last ranking company.  is a discrimination intensity function with non-negative and non-decreasing . The equation of  means that the weight value of  of  should be larger that the weight value  of . This paper applies  between the 1st to 10th rank,  between 11th to 25th,  between 26th to 35th, and  for 36th to 42nd (see Appendix Table 2), applying strong ordering [27, 43]. The result gives two 42 x 42 matrices for each company’s ranking results ( and a weight value of each company for each rank ( for all of i).

  1. Line 203: 42 excellence SCC companies or 40???

(Answer) We made changes as follows.

“…42 excellence SCC companies with two excluded companies, such as Inditex and Dell..”

Reviewer 2 Report

Thanks the Editor for the invitation to review this interesting paper.

The paper is well written using academic standards however I cannot find the relation of the topic and contents to Sustainability issue, so the Authors must consider how to improve the paper to meet sustainability aspects. Maybe following paper would be helpful:

Kot, S., ul Haque, A., & Kozlovski, E. (2019). Strategic SCM’s mediating effect on the sustainable operations: Multinational perspective. Organizacija, 52(3), 219-235.

Kot, S. (2018). Sustainable supply chain management in small and medium enterprises. Sustainability, 10(4), 1143.

Kot, S., Goldbach, I. R., & Ślusarczyk, B. (2018). Supply Chain Management in SMEs-Polish and Romanian Approach. Economics & Sociology, 11(4), 142.

Add information about time period of data collection

Conclusion is pure – add some more recommendations

Author Response

1. The paper is well written using academic standards however I cannot find the relation of the topic and contents to Sustainability issue, so the Authors must consider how to improve the paper to meet sustainability aspects. Maybe following paper would be helpful:

Kot, S., ul Haque, A., & Kozlovski, E. (2019). Strategic SCM’s mediating effect on the sustainable operations: Multinational perspective. Organizacija, 52(3), 219-235.

Kot, S. (2018). Sustainable supply chain management in small and medium enterprises. Sustainability, 10(4), 1143.

Kot, S., Goldbach, I. R., & Ślusarczyk, B. (2018). Supply Chain Management in SMEs-Polish and Romanian Approach. Economics & Sociology, 11(4), 142.

(Answer) We appreciate your recommendation of the valuable articles and cited Kot (2018) reflecting SCC with SHV to assure sustainable supply chain.

2. Add information about time period of data collection

(Answer) Thank you for the invaluable suggestions and comments. The time frame is from 2011 to 2017 (Line #14-16, 55-56, 197) for whole paper and preference voting method and from 2012 to 2016 for relationship analysis.

We changed as follows.

(Abstract, line #14 to 16)) The authors investigated nine FPIs that were selected from the financial statements of 155 companies within nine industries from 2011 to 2017 using Morningstar financial database and Gartner’s report.

(Introduction, line #55-56) Therefore, this research is to find the relationship between shareholder values and supply chain capability using companies’ financial statements from 2011 to 2017.

3. Conclusion is pure – add some more recommendations

(Answer) Many thanks for your comment. We read once again meticulously and changed appropriately.

Round 2

Reviewer 2 Report

Thank you for the revised version. I think Authors have done it properly.

the paper can be publushed,